# Pesticide Exposure in Relation to the Incidence of Abnormal Glucose Regulation: A Retrospective Cohort Study

**DOI:** 10.3390/ijerph19127550

**Published:** 2022-06-20

**Authors:** Sung-Kyung Kim, Hyun-Jung Oh, Sung-Soo Oh, Sang-Baek Koh

**Affiliations:** 1Department of Occupational and Environmental Medicine, Institute of Occupational and Environmental Medicine, Wonju College of Medicine, Yonsei University, Wonju 26426, Korea; stacte@yonsei.ac.kr (S.-K.K.); sy33kn@gmail.com (H.-J.O.); 2Department of Preventive Medicine, Wonju College of Medicine, Yonsei University, Wonju 26426, Korea

**Keywords:** pesticide exposure, diabetes mellitus, prediabetes, abnormal glucose regulation, longitudinal study, incidence, rural

## Abstract

Diabetes and prediabetes (called abnormal glucose regulation (AGR)) are adverse health effects associated with exposure to pesticides. However, there are few epidemiological studies on the relationship between pesticide use and the incidence of AGR. We examined the causal relationship between pesticide use and AGR incidence in a rural population using data from a Korean Farmers’ Cohort study of 1076 participants. Poisson regression with robust error variance was used to calculate the relative risks (RR) and 95% confidence intervals (CI) to estimate the relationship between pesticide exposure and AGR. The incidence of AGR in the pesticide-exposed group was 29.1%. Pesticide use increased the RR of AGR (RR 1.32, 95% CI 1.03–1.69). We observed a low-dose effect related to exposure of pesticides to AGR and a U-shaped dose–response relationship in men. Pesticide exposure is related to the incidence of AGR, and the causal relationship differs between men and women.

## 1. Introduction

Pesticides are toxic substances used to kill living organisms for various purposes [1], and agriculture is the largest industry that consumes pesticides (approximately 85% of global production) [2]. According to the Organization for Economic Cooperation and Development (OECD) data, Korea is third worldwide regarding agricultural chemical sales per hectare. Although agricultural production is increasing, the area of agricultural land in OECD countries continues to shrink [3]. However, this increase in agricultural production could have been caused by increasing the amount of pesticide input per unit area [4]. Some pesticides are known endocrine disruptors with a U-shaped dose–response relationship [5], and certain pesticides have health effects on the human body, including links to cognitive deficits, immune diseases, Parkinson’s disease, and decreased lung function. Dose–response associations with lung and prostate cancers have been reported in previous cross-sectional studies [6,7,8,9,10,11].

It is well known that diabetes increases the mortality rate of affected patients, causes complications when not properly treated, reduces the quality of life of patients, and causes socioeconomic problems [12,13]. The increased risk of coronary artery disease is already apparent in modestly elevated blood glucose levels, which are still below the present threshold for diabetes, such as prediabetes, including impaired fasting glucose (IFG) and impaired glucose tolerance (IGT) [14,15]. Diabetes and prediabetes are involved in abnormal glucose regulation (AGR) [16]. Newly detected abnormal glucose tolerance is one of the strongest prognostic factors after myocardial infarction (MI) [16] and acute ischemic stroke [17]. A prospective cohort study of 3450 people in China suggested that the incidence of ischemic stroke, intracranial hemorrhage, and subarachnoid hemorrhage (SAH) are associated with AGR [18]. As the risk of cardiovascular disease also increases in prediabetes, studies have shown that screening individuals for AGR can cost-effectively implement improved preventive measures for cardiovascular disease early [19,20].

Rapid urbanization and increasingly sedentary lifestyles in several countries have contributed in part to the increase in the prevalence of insulin-resistant diabetes [21]. Therefore, the risk factors for diabetes in the less urbanized rural population may differ from those mentioned above, and the possibility of pesticide exposure as a cause of diabetes and prediabetes in recent years has also been suggested. Animal experiments have suggested that exposure to organophosphorus pesticides induces insulin resistance [22], and in a prospective study of professional pesticide sprayers, it was found that the pesticide-use group was more likely to develop diabetes than the non-use group [23]. However, in this study, only days of cumulative use of pesticides and the presence or absence of self-reported diabetes were confirmed, without considering the condition of wearing protective equipment and the spraying method.

Therefore, our objective was to investigate the risk of AGR—including diabetes and prediabetes—using a detailed pesticide exposure questionnaire and population-based longitudinal data from the general population living in rural areas. Few studies have attempted to determine the causal relationship between cumulative exposure to pesticides and AGR, including prediabetes. Most previous studies on the association between pesticide exposure and diabetes, which did not include prediabetes, were cross-sectional studies. Additionally, previous studies that evaluated pesticide exposure by measuring pesticide metabolites by biomarker measurements are somewhat unreasonable for reflecting long-term chronic exposure to pesticides other than organochlorine pesticides that may remain in the human body [24]. Therefore, we attempted to calculate the levels of pesticide exposure of rural residents and examined the epidemiologic causality of the occurrence of AGR, including diabetes and prediabetes. In this study, impaired glucose regulation (IGR) refers to both IFG and IGT, while abnormal glucose regulation includes IGR and diabetes.

## 2. Materials and Methods

### 2.1. Study Population

This study used data from a cohort study of Korean farmers. This rural cohort study aimed to elucidate the prevalence, incidence, and risk factors of chronic diseases such as hypertension, diabetes, and cardiovascular disease. Participants lived in the rural areas of Wonju and Pyeongchang, Gangwon-do, Republic of Korea and were between 39 and 72 years of age [25]. The Institutional Review Board (IRB) of Yonsei University Wonju Severance Christian Hospital approved this study.

Baseline and follow-up examination were carried out between November 2005 and January 2008, and between April 2008 and August 2012, respectively. At baseline, 5178 subjects were chosen, of which 2568 participated in an additional survey of pesticide exposure. In this study, to determine the link between pesticide exposure and AGR, pesticide exposure or pesticide use was limited to occupational pesticide use, and did not include indirect exposure to the surrounding environment or pesticide ingestion through food or water. We excluded 217 participants with insufficient data on pesticide exposure, 603 participants who were lost to follow-up, and 672 participants with suspected diabetes or prediabetes detected on blood tests. Ultimately, 1076 participants were included in the study. The mean follow-up period was 2.64 years.

### 2.2. Covariates 

Continuous variables such as age, fasting blood glucose level, glycated hemoglobin (HbA1c), and body mass index (BMI) were expressed as mean ± standard deviation (SD), and categorical variables were expressed as counts with percentages (%). Smoking status was divided into three groups (non-smokers, former smokers, and current smokers). Alcohol use was divided into two groups according to current alcohol consumption (no/yes). Physical activity was divided into two groups according to regular exercise (no/yes). Patients were divided into two groups according to the monthly income of 1,500,000 Korean Won (<1,500,000 or ≥1,500,000), divided into two groups according to the highest educational qualification (elementary school graduation or lower, and middle school graduation or higher), and divided into two groups according to the marital status (married or other status). 

### 2.3. Data Collection

To collect data on pesticide exposure, we used a revised standardized questionnaire developed by the Agricultural Health Study (AHS) in a baseline survey. AHS, a study conducted by the National Cancer Institute, developed a quantitative method to estimate long-term pesticide exposure in a large prospective cohort study of more than 58,000 pesticide sprayers in North Carolina and Iowa [26]. In 2005, in a study by Coble et al., through a questionnaire, the pesticide exposure algorithm was found to provide a reasonably effective measure of the intensity of pesticide sprayer exposure compared to the results using biological monitoring [27]. Before this study, a pilot survey was conducted with a small sample of farmers living in Inje, Gangwon-do, Republic of Korea (*N* = 91). Data from this pilot sample were not used in subsequent analyses. The target respondents were adults between 39 and 72 living in rural areas. The pesticide exposure group was defined as those who sprayed pesticides occupationally (including those who owned farms or farm workers). The study was conducted through face-to-face interviews, after each participant provided written informed consent. The interviews were conducted in the local language, and participants were asked to provide detailed information on their use of pesticides. They were asked to state whether they had ever used pesticides and if they had mixed or applied any pesticides. The sum of years of pesticide use and the average number of days per year of pesticide use were also assessed. 

Although the types of pesticides were not considered in the questionnaire, among the possible pesticides to which our study subjects were exposed, the following pesticides have been reported to have a high probability of causing health effects; paraquat (quaternary ammonium herbicides), organophosphate insecticides, organophosphate herbicides, pyrethroid insecticides, carbamate insecticides, organochloride insecticides, and phenoxy herbicides [28]. In Korea, it has been reported that the production and sale of organochloride insecticides have been banned since the early 1970s, so their use and exposure are low, and there is a report that organophosphate insecticides are the most used [29]. Therefore, it was assumed that exposure to pesticides in the participants of this study was greater than that of the other pesticides listed above, except for organic chlorine pesticides.

### 2.4. Exposure Assessment

Exposure to pesticides can occur during transport, mixing, application of pesticides, or cleaning and repair of equipment. Based on these factors, the intensity of pesticide exposure and the cumulative exposure index (CEI) were calculated as follows [30]:*Intensity level = (mixing status + application method + repair status) × PPE score
CEI = intensity level × duration (number of years) × frequency (average number of days per year)

Pesticides were divided into three groups according to the mixing status (never mixed, mixed <50% of the instances, and mixed >50% of the instances; assigned 0, 3, and 9, respectively), which were further divided into six groups (does not apply, seed treatment or tablet distribution, backpack, hand spray, mist blower/fogger or airblast; assigned 0, 1, 8, and 9, respectively). The repair status was a two-level variable based on whether the equipment was repaired personally (0 and 2, respectively). For using PPE, participants were scored according to using protective equipment and divided into eight groups (allocated 1.0, 0.8, 0.7, 0.6, 0.5, 0.4, 0.3, and 0.1) [26]. Years of pesticide use (0, 1–12, and >12 years), frequency of pesticide use (0, 1–25, and >25 days average per year), scores (0, 0.3–0.6, <1, and 1), intensity level of pesticide exposure (0, ≤4, ≤9, ≤12, and >12), and CEI of pesticide exposure (0, ≤448, ≤2160, ≤5000, and >5000) were classified into four groups according to their quartile values.

The outcome was the incidence of AGR. The classification of AGR was based on standard cutoffs for FPG and HbA1c, as defined by ADA [31,32]. Normoglycemia was defined as a level of HbA1c < 5.7% and a level of FPG < 100 mg/dL. During the follow-up period, if the participant was diagnosed with diabetes by a doctor, if the fasting plasma glucose level was ≥126 mg/dL, or if the glycated hemoglobin level was ≥6.5%, it was defined as new-onset diabetes. Additionally, during the follow-up period, if the fasting blood glucose level was >100 mg/dL and <125 mg/dL, or if the HbA1c level was >5.7% and <6.4%, it was defined as new-onset prediabetes. 

### 2.5. Statistical Analysis

To examine the relationship between pesticide exposure and AGR, a *t*-test was initially used to calculate the number of participants and percentiles based on each continuous variable. A chi-square test was performed for each categorical variable to estimate the difference in the incidence of AGR according to the covariates and calculate the *p*-value. For common events such as the occurrence of AGR, the odds ratio can overestimate the relative risk (RR) [33]. Poisson regression, using a strong variance of error, is widely used to directly estimate the RR for both common and rare outcomes [25]. Therefore, in this study, the Poisson regression with strong error variance was used to calculate the RR and the 95% confidence interval (CI) to estimate the relationship between pesticide exposure and AGR. The RR was adjusted for age, sex, smoking status, alcohol consumption, BMI, regular exercise, monthly income, education, and marital status. Later, a stratified analysis by sex was performed. Statistical significance was established with a two-sided *p*-value of <0.05. All analyses were performed with the SPSS statistical software package (v21.0; IBM Co., Armonk, NY, USA).

## 3. Results

### 3.1. Descriptive Statistics of Variables

The descriptive and baseline information of the study population at enrollment is listed in Table 1. At enrollment, the mean fasting glucose was 87.1 mg/dL (SD ± 6.2) in the group without pesticide use and 87.5 mg/dL (SD ± 6.0) in the group with pesticide use, respectively (*p* = 0.32). For HbA1c, the mean fasting glucose was 5.21% (SD ± 0.01) in the group without pesticide use and 5.28% (SD ± 0.10) in the group with pesticide use (*p* <0.0001). There were also statistically significant differences in age and BMI between the groups without and with pesticide use. Furthermore, sex, smoking status, alcohol use, regular exercise, monthly income, education, and marital status were significantly different between groups.

### 3.2. Incidence of AGR According to Pesticide Exposure

AGR was observed in 23.1% and 29.1% of patients in the groups without pesticide use and with pesticide use, respectively (Table 2). The incidence of AGR according to other pesticide-related variables was not significant, except for the intensity level of pesticide exposure (*p* < 0.001). 

### 3.3. RRs of the Incidence of AGR by Variables Related to Pesticide Exposure 

The incidence of AGR was significantly higher in participants with pesticide use than in those with no pesticide use (Table 3). The incidence of AGR was 1.32 times higher in the pesticide exposure group than in non-pesticide exposure group, even after correcting for confounder variables (95% CI 1.03–1.69). The incidence of AGR was significantly elevated in study participants who had used a backpack or hand spray, even in the group with pesticide use for >30 years, and in the group with a PPE score between 0.1 and 0.8. These differences were statistically significant. All variables related to pesticide exposure were significantly associated with the incidence of AGR. When all confounder variables were corrected for each pesticide intensity, the group with intensity level of pesticide ≤4 was 1.80 times higher than that of the non-exposed group (95% CI 1.24–2.60). By the CEI pesticide group, only the group with ≥5000 showed 1.63 times higher than the nonexposed group (95% CI 1.18–2.27), compared to other CEI groups.

### 3.4. RRs of AGR Incidence by Variables Related to Pesticide Exposure in Male Participants 

In Table 4, the incidence RRs of AGR were significantly higher in the “ever used pesticides” group than in the reference group after adjustment for age, sex, smoking status, alcohol use, regular exercise, education and marital status (RR 1.81, 95% CI 1.22–2.67). The group of “<50% of the instances” (RR 2.27, 95% CI 1.27–4.07), seed treatment or tablet distribution (RR 2.77, 95% CI 1.48–5.20), “>30 years of pesticide use” (RR 3.03, 95% CI 1.74–5.27), 1–7 days of pesticide use per year (RR 2.14, 95% CI 1.28–3.55), pesticide exposure in a group of 4 score intensity (RR 2.01, 95% CI 1.10–3.68), and pesticide exposure at a highest CEI (RR 2.39, 95% CI 1.54–3.70) increased RR incidence most significantly in each category of variables related to pesticide exposure.

### 3.5. RRs of AGR Incidence by Variables Related to Pesticide Exposure in Female Participants 

Table 5 shows the RR for AGR in the female group, and some differences were observed between this group and the male group. The incidence RR of AGR was not significantly elevated in the group of “ever used pesticides” compared to that in the reference group after adjustment for age, sex, smoking status, alcohol use, regular exercise, education, and marital status (RR 1.10, 95% CI 0.80–1.51). Only pesticide exposure at a 12 intensity (RR 1.85, 95% CI 1.05–3.27) increased the RRs of incidence of AGR most significantly in each category of variables related to pesticide exposure. 

### 3.6. Dose–Response Patterns between Pesticide Exposure and Incidence of AGR 

Figure 1 and Figure 2 show the dose–response patterns between pesticide exposure and the incidence of AGR for all subjects and subgroups. Figure 1 shows the U-shaped dose–response pattern of AGR incidence as the intensity level of pesticide exposure increased. In particular, this was relatively clearer in the male group than in the female group. In the female group, the RR for AGR at a low intensity level was higher than in the non-pesticide exposed group, but the difference was not statistically significant. Figure 2 shows the relationship between the increase quartiles in CEI of pesticide exposure and RR of AGR incidence with a U-shaped dose–response pattern. Similarly to the case shown in Figure 1, the dose–response pattern was more clearly observed in male subjects than in female subjects.

## 4. Discussion

In this study, the occurrence of AGR in the pesticide use group was 29.1%, which was higher than 23.1% in the pesticide non-use group (*p*-value = 0.026). We also confirm the relationship between pesticide exposure and the resulting occurrence of AGR (RR 1.32, 95% CI 1.03–1.69). Furthermore, when CEI, an indicator of cumulative exposure to pesticides, was the first quartile, second quartile, third quartile, and fourth quartile, the RR were 1.46 (95% CI 0.93–2.31), 1.31 (95% CI 0.89–1.94), 1.29 (95% CI 0.85–1.95), and 1.63 (95% CI 1.18–2.27), respectively (Table 3). This indicates the possibility of low dose toxicity and a U-shaped dose–response relationship, as suggested in our previous study on the relationship between pesticide exposure and metabolic syndrome [25]. The RR of health effects in the low pesticide exposure group compared to the pesticide nonexposure group was higher than in the medium pesticide exposure group, and the pattern of increased RR again in the high pesticide exposure group becomes clearer in male participants with statistical significance when stratified by sex (in Table 4, the RRs are 2.45 (95% CI 1.45–4.12), 1.47 (95% CI 0.77–2.80), 1.80 (95% CI 1.06–3.07), and 1.88 (95% CI 1.18–3.01) in the first quartile, second quartile, third quartile, and fourth quartile, respectively). In female, unlike in male, the relationship between pesticide exposure and pesticide-related variables, and the occurrence of AGR was not significant, and the low dose toxicity and U-shaped dose–response relationships observed in Table 3 and Table 4 did not appear. This is believed to be because, as discussed in previous studies, there are potential differences in pesticide spraying behavior and biological differences [25]. This can be explained in detail by the differences in the roles of men and women in spraying pesticides. According to existing research in Korea, direct pesticide spraying is carried out primarily by male farmers, and female farmers intermittently assist in the spraying of pesticides, such as aligning and transporting pesticide lines when male farmers apply pesticides [11]. Therefore, it is possible that there is a difference between men and women in exposure to pesticides and the resulting health effects. Furthermore, differences in the metabolism and excretion of pesticides due to biological differences between men and women may also have influenced these results [34]. However, more studies are needed to clarify why the U-shaped dose–response relationship and low-dose toxicity between pesticide exposure and AGR are only particularly evident in the male group.

In our study, the incidence of AGR was 29.1% during a 2.64-year follow-up period. As most diabetic diseases occur after prediabetes [35], the short study period was inadequate to determine the occurrence of diabetes due to exposure to pesticides. However, this study is significant in that it is the first to address the occurrence of AGR according to the use and exposure of pesticides. A recent study found that diabetic patients had higher HbA1c levels when exposed to pesticides [36]. In a previous study, persistent organic pollutants (POPs) at low doses increased the risk of diabetes, while higher doses of POPs did not increase the risk [37]. In a study on the relationship between organophosphate exposure and neurological deficits [38], chlorpyrifos exposure and reprotoxicity in rats [39], lambda-cyhalothrin exposure and hepatotoxicity in rats [40], oxidative stress and inflammation induced by upregulation of inflammatory cytokines have been suggested as pathophysiological mechanisms. 

A study by Duzguner et al., which dealt with neonicotinoid pesticides and neurological and hepatotoxicity in rats, also suggested that oxidative stress and inflammation may participate in the development of diabetes [41]. Participants who used chlorpyrifos and diazinon were significantly more likely to develop diabetes [23], and organochlorine pesticides have been reported to be strongly associated with the probability of HbA1c levels >7% [42]. A meta-analysis of 11 cross-sectional studies and six prospective studies reported a general relative risk (95% CI) of organochlorine pesticides and type 2 diabetes of 2.30 (95% CI 1.81–2.93) [43]. Our present study is consistent with the results of these previous studies, but also presented the relative risk of diabetes or prediabetes according to sex and exposure to pesticides.

In a previous study, we confirmed that exposure to pesticides was related to the development of metabolic syndrome [25], and diagnostic criteria for metabolic syndrome included prediabetes and AGR status in fasting conditions such as diabetes [44]. Therefore, it is believed that the appearance of AGR caused by pesticides is related to an increase in oxidative stress caused by pesticides, similar to the appearance of a metabolic syndrome caused by pesticides. The fact that oxidative stress-induced insulin resistance is suggested to be a key mechanism in the metabolic syndrome caused by exposure to pesticides is consistent with the results of this study that oxidative stress-induced exposure to pesticides causes insulin resistance [25]. 

In this study, the risk of occurrence of AGR was shown to increase even when individuals were exposed to low concentrations of pesticides. Therefore, it is necessary to actively wear PPE even if the pesticide application period or frequency of application is small. This can also be improved through farmers’ pesticide education. For example, an agricultural council could encourage farmers to use pesticides responsibly by providing intensive training that includes information on the importance of using pesticides correctly, such as wearing protective gear and using pesticides in the right amount [45]. In addition to these efforts, doctors may consider prescribing drugs that improve insulin resistance, such as metformin. Metformin is a relatively safe and widely used drug for the treatment of type 2 diabetes mellitus. A systematic review in 2009 found that metformin reduced the rate of conversion of prediabetes to diabetes. Furthermore, according to a 2015 review article reviewing the effectiveness of metformin use in the prevention of diabetes for prediabetes status, it was concluded that metformin can be used to improve insulin resistance in prediabetes patients due to exposure to pesticides [46,47]. However, the timing of the introduction of metformin in prediabetes remains controversial, and more evidence is needed to improve long-term outcomes in patients with metformin-treated prediabetes [47]. Furthermore, a 2020 study by Davison et al. found that metformin use to treat prediabetes is likely to significantly increase individual drug costs. Therefore, lifestyle interventions, particularly weight loss in overweight and obese people, are more likely to be pursued than medication use [48]. Therefore, even in prediabetic patients exposed to pesticides, lifestyle changes may be a better choice than metformin treatment.

Conversely, in a situation where the prevalence of diabetes and the burden thereof increase, prevention of new diabetes and prediabetes caused by exposure to pesticides may reduce some of these burdens. The nanotechnology related to pesticides is expected to play an important role. One of the goals of nanotechnology in agriculture is to reduce the use of toxic chemicals. Incorporated nanoformulations with herbicides have also been reported to reduce the environmental toxicity and genotoxic effects of agricultural products. As a result, the use of these nanoformulations can reduce the farmer’s cumulative exposure to pesticides and can be expected to have a preventive effect on various health effects, such as decreased insulin function due to exposure to pesticides [49].

Our study has some limitations. First, total pesticide exposure was calculated using the method described by Dosemeci et al. [26]. Although this method has been validated in previous studies, it has the disadvantage of being unable to assess the health effects of individual pesticides [25]. However, since one component of pesticides exhibits various toxic effects in the human body and different types of pesticides have the same toxicity, it is still difficult to classify individual pesticides to accurately evaluate their health effects [50]. Therefore, to calculate the amount of exposure for each pesticide type, a new exposure assessment tool should be developed according to the assumption that it can be improved over the existing exposure assessment method in the future. Second, although only occupational pesticide exposure was calculated in this study, there is a limitation that pesticide exposure due to food, beverages, or other environmental exposures was not calculated. Third, due to the relatively short follow-up period, the incidence of diabetes was very low. Therefore, it was difficult to calculate the RR. However, because the risk of the occurrence of AGR found in the pesticide exposure group increased significantly during the short period of adhesion, the long-term relationship between pesticide exposure and the occurrence of AGR could be compared with the results of the present study in the future. Finally, plasma glucose level 2 h after a 75 g oral glucose load test (OGTT) is also one of the indicators used for measuring insulin resistance, such as impaired glucose tolerance and the diagnosis of diabetic diseases. However, it was not available in this study, so it was not used for statistical analysis. However, according to a study by Knudsen et al., AGR classification using early OGTT did not provide reliable information on the long-term glucometabolic state, and HbA1c and fasting plasma glucose measured in the hospital were more useful markers of glucometabolic disturbance [51]. However, because OGTT was not measured as an indicator of AGR, there is the possibility of misclassification in some study participants who were normal in fasting blood glucose or glycated hemoglobin, but may be abnormal in the definition of AGR by OGTT. Additionally, there is a possibility that type 1 diabetes may have developed among study subjects newly diagnosed with AGR. However, this study could not exclude this, so it is possible that the relative risk of occurrence of AGR was estimated to be rather large. Moreover, data on the subjects’ past history of chronic diseases—such as cardiovascular disease or cerebrovascular disease—which can affect diabetes, and data on variables such as usual diet, familial history, and genetic factors have not been collected. Furthermore, the inability to determine the incidence of cardiovascular diseases that can affect the occurrence of diabetes may have influenced the value of RR for the occurrence of AGR. Finally, the pesticide exposure questionnaire should be structured in more detail. The questionnaire used in this study has proven reliability and validity, but since it has been developed for nearly 20 years, it may need to be renewed to reflect the latest trends in agrochemical research. In future research, sociodemographic issues, phytosanitary assessment, cancer risk, knowledge, and behavior of farmers with respect to the use of pesticides should be reflected in the questionnaire to conduct a study related to pesticide exposure [45]. 

Despite these limitations, our study had several advantages. This is the first study to consider the causal link between pesticide use and AGR, including diabetes and prediabetes. Additionally, to our knowledge, the risk of AGR occurrence was calculated in relative detail according to the level of pesticide exposure. Based on this, the possibility of a low-dose toxic action as a mechanism for the occurrence of AGR by exposure to pesticides was presented for the first time in an epidemiological study. Finally, we included more than 1000 participants in the analyses, which is one of the strengths of this cohort study. On the contrary, most of the studies evaluating pesticide exposure by measuring serum biomarkers were small or cross-sectional. There were prospective studies that did not include pre-diabetes and self-reported incident diabetes as the primary outcome. Therefore, we think that our study has the advantage of defining the status of AGR through blood tests [24].

## 5. Conclusions

Exposure to pesticides can lead to more pronounced health effects related to insulin resistance in men than in women. Although several well-known factors affect insulin resistance in adults, exposure to pesticides may be related to the pathophysiology of diabetes. Therefore, it may be helpful to emphasize reducing pesticide exposure as much as possible, especially in diabetes prevention education for farmers or people living in countries with high pesticide use.

## Figures and Tables

**Figure 1 ijerph-19-07550-f001:**
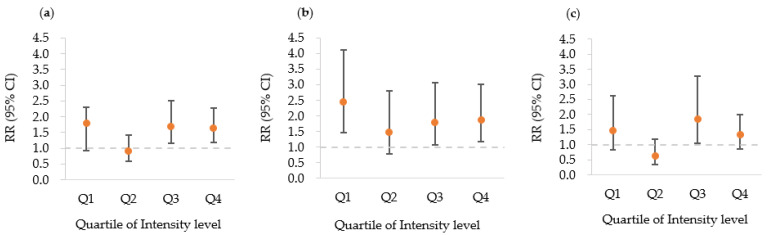
Relationships between quartiles increase in the intensity level of pesticide exposure and the relative risk (RR) of the incidence of AGR, in the (**a**) total subjects, (**b**) male subjects, and (**c**) female subjects. The RRs were adjusted for age, smoking status, alcohol use, BMI, regular exercise, education, monthly income, and marital status. The error bars represent the 95% confidence interval for each estimate of points. The quartile values for each intensity level are provided in Table 2. CI, confidence interval.

**Figure 2 ijerph-19-07550-f002:**
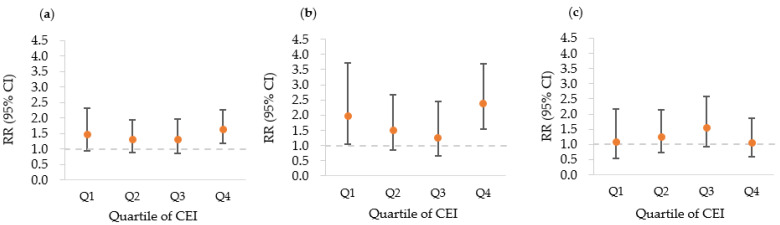
Relationships between the quartiles increase in the cumulative exposure index (CEI) of pesticide exposure and the relative risk (RR) of the incidence of AGR, in the (**a**) total subjects, (**b**) male subjects, and (**c**) female subjects. The RRs were adjusted for age, smoking status, alcohol use, BMI, regular exercise, education, monthly income, and marital status. The error bars represent the 95% confidence interval for each estimate of points. The quartile values for each CEI are provided in Table 2. CI, confidence interval.

**Table 1 ijerph-19-07550-t001:** Baseline demographic characteristics of participants according to pesticide use (*n* = 1076).

Demographic Characteristics	Pesticide Use		*p*-Value
	No (*n* = 660)	Yes (*n* = 502)	
	Mean (±SD) * or Frequency (%) **		
Sex			
Male	159 (27.0%)	236 (48.4%)	<0.0001
Female	429 (73.0%)	252 (51.6%)	
Age (years)	52.8 (±8.1)	54.7 (±8.0)	<0.0001
Fasting glucose (mg/dL)	87.1 (±6.2)	87.5 (±6.0)	0.32
HbA1c (%)	5.21 (±0.01)	5.28 (±0.10)	<0.0001
BMI (kg/m^2^)	23.9 (±2.9)	24.3 (±3.1)	0.02
Smoking status			
Non-smoker	485 (82.8%)	338 (69.8%)	<0.0001
Ex-smoker	41 (7.0%)	65 (13.4%)	
Current smoker	60 (10.2%)	81 (16.8%)	
Alcohol use			
No	377 (64.4%)	256 (52.6%)	<0.0001
Yes	208 (35.6%)	231 (47.4%)	
Regular exercise			
No	354 (60.7%)	416 (86.0%)	<0.0001
Yes	229 (39.3%)	68 (14.0%)	
Monthly income (Korean won)			
<1,500,000	255 (47.2%)	312 (74.6%)	<0.0001
≥1,500,000	285 (52.8%)	106 (25.4%)	
Education			
Elementary school or below	207 (35.3%)	318 (65.3%)	<0.0001
Middle school or higher	379 (64.7%)	169 (34.7%)	
Marital status			
Married	511 (87.4%)	450 (93.6%)	<0.001
Others	74 (12.6%)	31 (6.4%)	

Abbreviations: BMI, body mass index; HbA1c. glycated hemoglobin; SD, standard deviation. * *p*-value from the *t*-test, ** *p*-value from the chi-square test.

**Table 2 ijerph-19-07550-t002:** Incidence of AGR according to pesticide exposure (*n* = 1076).

Pesticide-Related Variables	Incidence of AGR, *n* (%)	*p*-Value *
Pesticide use		0.026
No	136 (23.1%)	
Yes	142 (29.1%)	
Pesticide mixing status		0.238
No	176 (24.3%)	
<50% of the instances	38 (30.7%)	
>50% of the instances	64 (28.0%)	
Application method		0.062
No	136 (23.1%)	
Seed treatment or tablet distribution	30 (25.2%)	
Backpack	18 (37.5%)	
Hand spray	58 (31.7%)	
Mist blower/fogger or air-blast	36 (26.1%)	
Years of pesticide use		0.104
0	136 (23.1%)	
≤15	28 (27.5%)	
≤25	34 (25.0%)	
≤30	29 (27.9%)	
>30	39 (35.1%)	
Frequency of pesticide use (per year)		0.092
0	136 (23.1%)	
≤7	39 (35.5%)	
≤15	28 (27.5%)	
≤20	29 (25.4%)	
>20	30 (28.0%)	
Scores for PPE		0.140
0	136 (23.1%)	
≤0.6	37 (28.0%)	
<1	16 (26.2%)	
1	89 (30.2%)	
Intensity level of pesticide exposure		<0.001
0	136 (23.1%)	
≤4	25 (39.7%)	
≤9	27 (20.3%)	
≤12	29 (34.5%)	
>12	61 (29.3%)	
CEI of pesticide exposure		0.201
0	136 (23.1%)	
≤448	16 (34.0%)	
≤2160	32 (28.3%)	
≤5000	28 (25.9%)	
>5000	48 (30.4%)	

Abbreviations: CEI, cumulative exposure; PPE, personal protective equipment; AGR, abnormal glucose regulation. * *p*-value from the chi-square test.

**Table 3 ijerph-19-07550-t003:** RRs of AGR related to pesticide exposure.

Pesticide-Related Variables	Crude RR (95% CI)	RR (95% CI) *	RR (95% CI) **
Pesticide use			
No	1 (reference)	1 (reference)	1 (reference)
Yes	**1.26 (1.03–1.54)**	**1.25 (1.02–1.53)**	**1.32 (1.03–1.69)**
Pesticide mixing status			
No	1 (reference)	1 (reference)	1 (reference)
<50% of the instances	**1.35 (1.00–1.83)**	1.35 (0.99–1.84)	1.33 (0.92–1.91)
>50% of the instances	1.16 (0.91–1.48)	1.20 (0.93–1.56)	1.30 (0.98–1.74)
Application method			
No	1 (reference)	1 (reference)	1 (reference)
Seed treatment or tablet distribution	1.08 (0.77–1.52)	1.03 (0.73–1.47)	1.14 (0.75–1.73)
Backpack	**1.61 (1.09–2.39)**	**1.72 (1.14–2.58)**	**1.74 (1.13–2.66)**
Hand spray	**1.41 (1.08–1.83)**	**1.46 (1.10–1.92)**	**1.55 (1.12–2.15)**
Mist blower/fogger or air-blast	**1.59 (1.08–2.34)**	1.20 (0.86–1.67)	1.36 (0.94–1.95)
Years of pesticide use			
0	1 (reference)	1 (reference)	1 (reference)
≤15	1.22 (0.86–1.74)	1.24 (0.87–1.77)	1.26 (0.85–1.87)
≤25	1.10 (0.79–1.54)	1.17 (0.84 –1.64)	1.35 (0.93–1.97)
≤30	1.24 (0.87–1.75)	1.23 (0.87–1.76)	1.40 (0.92–2.12)
>30	**1.55 (1.15–2.08)**	**1.45 (1.07–1.98)**	**1.69 (1.17–2.45)**
Frequency of pesticide use (per year)			
0	1 (reference)	1 (reference)	1 (reference)
≤7	**1.60 (1.19–2.16)**	**1.51 (1.11–2.06)**	**1.57 (1.10–2.26)**
≤15	1.18 (0.83–1.67)	1.18 (0.83–1.68)	1.35 (0.92–1.97)
≤20	1.17 (0.82–1.67)	1.16 (0.81–1.65)	1.31 (0.86–1.98)
>20	1.20 (0.86–1.69)	1.27 (0.90–1.81)	1.41 (0.97–2.06)
Scores for PPE			
0	1 (reference)	1 (reference)	1 (reference)
0.1–0.8	1.22 (0.93–1.61)	1.24 (0.93–1.65)	**1.41 (1.02–1.93)**
1	**1.32 (1.05–1.67)**	**1.30 (1.03–1.64)**	**1.39 (1.06–1.84)**
Intensity level of pesticide exposure			
0	1 (reference)	1 (reference)	1 (reference)
≤4	**1.70 (1.21–2.39)**	**1.72 (1.22–2.43)**	**1.80 (1.24–2.60)**
≤9	0.90 (0.62–1.31)	0.90 (0.62–1.31)	0.92 (0.59–1.42)
≤12	**1.53 (1.10–2.14)**	**1.58 (1.13–2.23)**	**1.69 (1.15–2.50)**
>12	**1.30 (1.00–1.69)**	**1.28 (0.98–1.67)**	**1.48 (1.08–2.02)**
CEI of pesticide exposure			
0	1 (reference)	1 (reference)	1 (reference)
≤448	1.46 (0.96–2.23)	1.42 (0.91–2.20)	1.46 (0.93–2.31)
≤2160	1.29 (0.92–1.81)	1.29 (0.92–1.81)	1.31 (0.89–1.94)
≤5000	1.11 (0.78–1.58)	1.14 (0.80–1.64)	1.29 (0.85–1.95)
>5000	**1.36 (1.02–1.80)**	**1.36 (1.02–1.82)**	**1.63 (1.18–2.27)**

Abbreviations: RR, relative risk; CI, confidence interval. * Adjusted for age and sex. ** Adjusted for age, smoking status, alcohol use, BMI, regular exercise, education, and marital status. The bold text shows the statistical significance of the RRs.

**Table 4 ijerph-19-07550-t004:** AGR RRs of AGR related to exposure to pesticides for men.

Pesticide-Related Variables	Crude RR (95% CI)	RR (95% CI) *	RR (95% CI) **
Pesticide use			
No	1 (reference)	1 (reference)	1 (reference)
Yes	**1.45 (1.01–2.1)**	**1.44 (1.00–2.08)**	**1.81 (1.22–2.67)**
Mixing status of pesticide			
No	1 (reference)	1 (reference)	1 (reference)
<50% of the instances	1.39 (0.88–2.20)	1.37 (0.87–2.16)	**1.69 (1.04–2.76)**
>50% of the instances	1.31 (0.90–1.91)	1.30 (0.89–1.90)	**1.66 (1.11–2.47)**
Application method			
No	1 (reference)	1 (reference)	1 (reference)
Seed treatment or tablet distribution	**2.48 (1.16–5.31)**	**2.48 (1.15–5.34)**	**2.77 (1.48–5.20)**
Backpack	**1.86 (1.12–3.09)**	**1.83 (1.11–3.01)**	**2.26 (1.35–3.78)**
Hand spray	1.44 (0.93–2.21)	1.42 (0.92–2.19)	**1.75 (1.10–2.78)**
Mist blower/fogger or air-blaster	1.21 (0.73–2.01)	1.21 (0.73–2.00)	1.61 (0.93–2.79)
Years of pesticide use			
0	1 (reference)	1 (reference)	1 (reference)
≤15	1.09 (0.60–2.01)	1.10 (0.60–2.02)	1.19 (0.63–2.25)
≤25	1.04 (0.59–1.85)	1.05 (0.59–1.86)	1.64 (0.95–2.81)
≤30	**1.63 (1.01–2.63)**	**1.62 (1.00–2.62)**	**2.45 (1.46–4.12)**
>30	**2.02 (1.30–3.14)**	**1.97 (1.26–3.09)**	**3.03 (1.74–5.27)**
Frequency of pesticide use (per year)			
0	1 (reference)	1 (reference)	1 (reference)
≤7	**1.93 (1.21–3.08)**	**1.88 (1.17–3.02)**	**2.14 (1.28–3.55)**
≤15	1.18 (0.68–2.05)	1.16 (0.67–2.02)	1.52 (0.86–2.67)
≤20	1.50 (0.87–2.58)	1.48 (0.86–2.54)	**2.01 (1.09–3.71)**
>20	1.44 (0.89–2.33)	1.45 (0.90–2.35)	**1.86 (1.15–3.02)**
Scores for PPE			
0	1 (reference)	1 (reference)	1 (reference)
0.1–0.8	**2.14 (1.29–3.55)**	**2.01 (1.09–3.71)**	1.54 (0.98–2.44)
1	1.52 (0.86–2.67)	**1.87 (1.15–3.02)**	1.30 (0.70–2.41)
Intensity level of pesticide exposure			
0	1 (reference)	1 (reference)	1 (reference)
≤4	**1.96 (1.19–3.24)**	**1.94 (1.17–3.20)**	**2.45 (1.45–4.12)**
≤9	1.20 (0.67–2.15)	1.18 (0.66–2.11)	1.47 (0.77–2.80)
≤12	1.52 (0.90–2.57)	1.52 (0.91–2.56)	**1.80 (1.06–3.07)**
>12	1.42 (0.91–2.22)	1.40 (0.89–2.20)	**1.88 (1.18–3.01)**
CEI of pesticide exposure			
0	1 (reference)	1 (reference)	1 (reference)
≤448	1.72 (0.93–3.17)	1.73 (0.93–3.19)	**1.96 (1.04–3.71)**
≤2160	1.51 (0.91–2.50)	1.48 (0.89–2.45)	1.50 (0.85–2.63)
≤5000	0.83 (0.44–1.54)	0.83 (0.44–1.54)	1.26 (0.65–2.44)
>5000	**1.90 (1.24–2.91)**	**1.88 (1.23–2.88)**	**2.39 (1.54–3.70)**

* Adjusted for age. ** Adjusted for age, smoking status, alcohol use, BMI, regular exercise, monthly income, education, and marital status. The bold text shows the statistical significance of the RRs.

**Table 5 ijerph-19-07550-t005:** RRs of AGR related to pesticide exposure for women.

Pesticide-Related Variables	Crude RR (95% CI)	RR (95% CI) *	RR (95% CI) **
Pesticide use			
No	1 (reference)	1 (reference)	1 (reference)
Yes	1.19 (0.93–1.54)	1.16 (0.89–1.49)	1.10 (0.80–1.51)
Mixing status of pesticide			
No	1 (reference)	1 (reference)	1 (reference)
<50% of the instances	1.44 (0.93–2.23)	1.37 (0.89–2.12)	1.11 (0.63–1.95)
>50% of the instances	1.10 (0.76–1.60)	1.12 (0.77–1.62)	1.17 (0.75–1.81)
Application method			
No	1 (reference)	1 (reference)	1 (reference)
Seed treatment or tablet distribution	0.96 (0.66–1.39)	0.94 (0.65–1.37)	0.92 (0.57–1.46)
Backpack	1.53 (0.62–3.81)	1.52 (0.61–3.77)	1.11 (0.36–3.41)
Hand spray	**1.60 (1.12–2.28)**	**1.54 (1.07–2.22)**	1.50 (0.94–2.40)
Mist blower/fogger or air-blaster	1.28 (0.82–1.99)	1.19 (0.77–1.85)	1.25 (0.76–2.04)
Years of pesticide use			
0	1 (reference)	1 (reference)	1 (reference)
≤15	1.41 (0.91–2.19)	1.36 (0.89–2.07)	1.31 (0.83–2.07)
≤25	1.21 (0.81–1.83)	1.31 (0.87–1.97)	1.34 (0.82–2.18)
≤30	0.98 (0.54–1.78)	0.91 (0.50–1.66)	0.79 (0.35–1.80)
>30	1.22 (0.76–1.95)	1.08 (0.66–1.77)	1.10 (0.60–2.02)
Frequency of pesticide use (per year)			
0	1 (reference)	1 (reference)	1 (reference)
≤7	1.47 (0.97–2.23)	1.32 (0.86–2.02)	1.26 (0.77–2.08)
≤15	1.33 (0.84–2.11)	1.31 (0.82–2.09)	1.40 (0.83–2.35)
≤20	1.00 (0.61–1.67)	0.98 (0.60–1.60)	0.93 (0.49–1.77)
>20	1.08 (0.62–1.88)	1.15 (0.65–2.02)	1.09 (0.54–2.17)
Scores for PPE			
0	1 (reference)	1 (reference)	1 (reference)
0.1–0.8	0.99 (0.61–1.60)	0.99 (0.61–1.61)	1.23 (0.73–2.05)
1	1.63 (0.74–3.61)	1.54 (0.74–3.22)	1.40 (0.59–3.31)
Intensity level of pesticide exposure			
0	1 (reference)	1 (reference)	1 (reference)
≤4	1.63 (0.98–2.72)	1.56 (0.94–2.59)	1.46 (0.82–2.61)
≤9	0.76 (0.46–1.26)	0.75 (0.45–1.24)	0.63 (0.33–1.19)
≤12	**1.75 (1.11–2.77)**	**1.74 (1.10–2.75)**	**1.85 (1.05–3.27)**
>12	1.29 (0.93–1.80)	1.22 (0.88–1.71)	1.32 (0.87–1.99)
CEI of pesticide exposure			
0	1 (reference)	1 (reference)	1 (reference)
≤448	1.36 (0.73–2.55)	1.14 (0.59–2.23)	1.09 (0.55–2.17)
≤2160	1.22 (0.75–1.99)	1.23 (0.76–1.98)	1.25 (0.73–2.14)
≤5000	**1.65 (1.10–2.48)**	**1.63 (1.09–2.43)**	1.54 (0.93–2.57)
>5000	1.01 (0.64–1.58)	0.99 (0.63–1.55)	1.06 (0.60–1.85)

* Adjusted for age. ** Adjusted for age, smoking status, alcohol use, BMI, regular exercise, education, monthly income, and marital status. The bold text shows the statistical significance of the RRs.

## Data Availability

Not applicable.

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
