# Peer review of "Pesticide Exposure in Relation to the Incidence of Abnormal Glucose Regulation: A Retrospective Cohort Study"

_ijerph, 2022, doi:10.3390/ijerph19127550_

Round 1
Reviewer 1 Report
Many results, graphical part is totally missing (maybe the authors will find a way to fulfil this aspect as well), and please see below my suggestions/requests regarding this manuscript:
L60-64. Please make the aim of the study a separate, last paragraph of Introduction section (to make it easier visible for those interested in the topic), highlighting better some aspects by responding to the following questions: Which is the novelty of your study or the special aspects it brings to the field? What makes different your study from others in the same/similar topic, already published? - Related to L 299-300 from Conclusions section
2nd section. Nothing about questionnaire detailing. A new subsection must be added, presenting all information about the surveys: who made/conceived the surveys? who validated them? there were some collaborations with sociologists, specialists in such questionnaires? were these questionnaires pre-tested before their application to all respondents? based on which criteria the items in the surveys were chosen/ how do you have chosen/decided the optimal items? based on which criteria, the respondents were chosen? etc. I suggest checking and referring to Ben Khadda, Z.; Fagroud, M.; El Karmoudi, Y.; Ezrari, S.; Berni, I.; De Broe, M.; et al. Farmers’ Knowledge, Attitudes, and Perceptions Regarding Carcinogenic Pesticides in Fez Meknes Region (Morocco). Int. J. Environ. Res. Public Health 2021, 18, 10879. https://doi.org/10.3390/ijerph182010879 , this paper will help also regarding the "pesticide" info.
Also in the Methodology, please better detail to what potential pesticides have the subjects been exposed. Could you also detail if patients developed T1DM or T2DM? Did you observe increased CV morbidity in the pesticide exposure group?
L129. As the unit of measure for volume in your research, please replace dl with dL (as Litter being the international unit of measure for volume). Please check/revise the entire manuscript in this regard.
Discussion. Following ideas must be completed:
- Please improve this section and give examples of how to minimize pesticide exposure in order to reduce diabetes risk.
- If pesticides cause insulin resistance, could metformin use in patients with AGR prevents the progression towards diabetes? Detail please.
- Moreover, what is the trend in using nanotechnology to optimally address the issue of pesticide toxicity (I suggest checking and referring to Behl et al. The dichotomy of nanotechnology as the cutting edge of agriculture: Nano-farming as an asset versus nanotoxicity, Chemosphere 2022, 288 Part 2, 132533. https://doi.org/10.1016/j.chemosphere.2021.132533
- After L 297 and completing I suggested above, please add a paragraph describing the strengths and the limitations of your study.
Reviewer 2 Report
This study is well-designed and written. Recently, a lot of papers in the field of "Pesticide exposure and diabetes mellitus" were reported. The author should renew these information and provide these information in the introduction and discussion. In addition, the author should illustrate what are the significances and differences in their study compared to other groups.
Reviewer 3 Report
This study investigated the causal relationship between pesticide use and AGR incidence in a rural population using data from a Korean farmers cohort study.
Before publication, the following parts need to be reinforced.
1. As can be seen from the following link, there was a report about the correlation between pesticide use and diabetes development several years ago.
https://www.webmd.com/diabetes/news/20150916/pesticide-exposure-tied-to-diabetes-risk
If this study is different from previous findings, reinforcement is needed in terms of novelty.
2. Diabetes has a variety of pathogenesis. It is necessary to analyze the population, including information on the presence of chronic diseases, usual diet, and genetic factors.
3. It is necessary to discuss the reason for the gender difference in the relationship between pesticide exposure and AGR incidence.
4. Tables 4 and 5 contain the key conclusions of this paper. It is recommended to express the values ​​in the Tables corresponding to the take-home message as a graph (e.g., bar graph).
minor point
the following statement should be corrected.
page 10, line 291: However, Knudsen et. Al. According to a study by et al., .....
Round 2
Reviewer 1 Report
The authors responded to all my suggestions.
Reviewer 3 Report
The manuscript has been significantly improved. It is recommended to be published in this journal.